# Pilot Tests on the Treatment of Bath Wastewater by a Membrane Bioreactor

**DOI:** 10.3390/membranes11020085

**Published:** 2021-01-25

**Authors:** Yan Shi, Songtao Zhong, Zhaohui Li

**Affiliations:** 1School of Environmental and Municipal Engineering, North China University of Water Resources and Electric Power, Zhengzhou 450045, China; zstt98@163.com; 2Department of Geosciences, University of Wisconsin—Parkside, 900 Wood Road, Kenosha, WI 53144, USA

**Keywords:** MBR, concentrating water areas, bath wastewater, wastewater recycling

## Abstract

In order to save water and reduce the cost of water in public areas, we studied the feasibility of recycling bath wastewater by a membrane bioreactor (MBR) at a college campus scale. The results showed that the treatment of bath wastewater by the MBR could achieve a chemical oxygen demand (COD) <50 mg/L with an average removal rate of 83%, a final NH_3_–N concentration of <10 mg/L with an average removal rate of 72%, and a turbidity of <0.5 ntu, with an average value of chromaticity of 26.4 tu. The treated water could meet or exceed the urban miscellaneous water standard of China (GB/T 18920-2002), and the processing cost is 1.70 CNY/m^3^ or 0.25 USD/m^3^, which is below the price of tap water. The results demonstrated both the economic benefit and the conservation of natural resources.

## 1. Introduction

Along with social, economic and demographic development, water demand has increased drastically. The supply and demand contradiction intensifies day by day in some countries [1,2]. In addition, water pollution aggravates the shortage of water resources. In order to realize sustainable use of water resources, saving water [3,4], water resources management [2,5,6,7,8,9] and sewage treatment technology [10,11,12] are researched to improve the circulation of water resources utilization.

On average, bath water usage is about 0.4 tons per adult per use in China. Considering the population of China, bathwater consumption is becoming an important issue in water resources management. One of the advantages of recycling bath wastewater is that the water is relatively stable and easy to be collected in all kinds of drainage water. In recent years, the transformation of many universities in China from elite education to mass education has resulted in a vast expansion in the scales of colleges and universities. As such, sewage discharge is also growing. Among sewage in universities, bath wastewater accounts for about 30%. If bath wastewater could be recycled, this could not only avoid wasting precious water resources, but also reduce the high cost of water bills for universities. Therefore, recycling bath wastewater in university campuses as a research object is of great importance. At present, different physical treatments [13,14], chemical treatments [15,16], biological treatments [17,18,19], membrane methods and composite methods [20,21] have been adopted to treat bathing wastewater, such as coagulation, filtration, adsorption [22,23], membrane filters [24,25,26,27], electrocoagulation–flotation [28] and UV disinfection [29].

In recent years, membrane separation technology has been introduced into biological wastewater treatment systems. A membrane bioreactor (MBR) is a device that combines a membrane module with a biological reactor to form a biochemical reaction system. MBRs can be divided into three types: separated, single piece and gravity submerged. In order to save water and improve water use efficiency, MBR technology was chosen as an important technology for the engineering of bath wastewater recycling [30,31,32]. Wang et al. researched a nonwoven membrane bioreactor for bath wastewater [33]. Zhou et al. conducted an experimental study on enhanced adsorptive biomembrane for the treatment of bathing sewage [34]. Wang et al. had an experimental study on the “Treatment of Bath Wastewater by Coagulation-AF-BAF Combination Process” [35]. According to the relative position of the membrane module and the bioreactor, a membrane bioreactor can be divided into a separated membrane bioreactor, an integral membrane bioreactor and a composite membrane bioreactor. Although the principles of membrane separation were determined about 250 years ago, full industrial application of membrane separation technology was not started until the 1960s. From reverse osmosis technology in the 1960s to the pervaporation technology of the 1990s, membrane separation technology has been rapidly developing. Especially after the 1990s, with the development of the composite membrane, the application domain of membrane separation technology expanded significantly and played a huge role in the water processing industry [36,37]. With the constant increase in water usage, bath wastewater recycling has attracted more attention.

Membrane separation technology has many advantages, such as its high separation efficiency, simple equipment, ease of operation, lack of phase changes and energy conservation. Thus, it has great potential application in the field of wastewater reuse. However, the cost of the membrane is relatively high, and the problem of membrane fouling may shorten the service life of the membrane device. We need to research membrane technology further for its future large-scale application. In this study, we evaluated the use of an MBR made of polypropylene at the pilot scale to assess the efficiency of the MBR and the water quality of bath wastewater after treatment.

## 2. Test Device and Method

### 2.1. Wastewater Source and Water Quality

The bath wastewater used came from a lady’s bath sewage water well in North China University of Water Resources and Electric Power, Zhengzhou, China. The main pollutants in the bathing wastewater are the secretion of human skin, grease, dander, hair, dirt, synthetic detergent and perfume, as well as bacteria, fungi, *E. coli* and viruses.

A submersible sewage pump was used to pump the wastewater into a storage tank located at a higher elevation, whose water level was gauge-controlled automatically. The water quality of the original bath wastewater is shown in Table 1 [38].

### 2.2. Testing Device and Procedure

The MBR reactor was made of a glass container with a dimension of 1.25 m by 1.0 m by 1.0 m (Figure 1). It has an effective water storage volume of 1 m^3^. A curtain box membrane, made of polypropylene hollow fibers and manufactured by KaiHua membrane material in Zhejiang University, was installed in the container. The hollow fibers have some advantages, such as their high packing density, harmlessness, anti-pollution nature and small footprint. The membrane had a pore size of 0.2 μm, a porosity of 50% and a total area of 16 m^2^. Two monolithic three-tier slices were assembled in parallel. The membrane was placed 0.3 m from the bottom and side walls with a spacing of 0.4 m in between. The test procedure was sketched in Figure 2.

## 3. Initial Experimental Section

### 3.1. Process of Inoculation

The important factors are (1) the strains from similar sewage processing factories, and (2) the strains from sewage properties similar to those of sewage plants [39,40,41,42]. These factors were considered to ensure the treatment capacity of the membrane, to ensure uniform contact between the membrane and the sewage, and uniform aeration. The handling process in Zhengzhou Wang Xin-Zhuang Sewage Treatment Plant was a conventional activated sludge process. The main source of wastewater was sewage, with a small amount of industrial wastewater. Thus, the sewage treatment process and wastewater quality in the plant and the pilot test were similar, which conforms to the principle of sludge inoculation. The sludge returning from the second pond in the plant was used as the screening sludge.

### 3.2. Methods of Cultivation for Activated Sludge

Both inoculation and stuffy exposure (that is, the batch culture method) were used for sludge cultivation, which was divided into three stages.

Complete stuffy exposure [43] was used in the first stage. The volumetric gas:water ratio was controlled between 20:1 and 25:1, and air flow was maintained at about 2.5 m^3^/h. Because of the sludge tender, the discrete exposure volume should not be too big, which was controlled at 1/2 of the normal design (about 1.7 m^3^/h), so as to facilitate formation of flocs. This stage was operated for 3 d.

The second stage was the buffering. The water was continuously pumped during the day, and completely stuffy exposed at night. The aeration rate was controlled at about 2.5 m^3^/h. At this stage, the microbial quantity was maintained at 6000–7600 mg/L (mixed liquor suspended solids (MLSS)), which did not meet the requirements at that point. This stage was operated for 14 d.

The third stage began with the operation of cultivation continuously, while the stuffy exposure was discontinued. To meet the needs of microbial growth, it is necessary to maintain a definite ratio of food to microbes (F/M ratio), and to keep the BOD5:N:P at 100:5:1, by coordinating a total of four additives used for the process to meet the requirements of the microbial carbon source. At this stage, the microorganisms grew faster, and the number met the design requirements of 3000 mg/L (MLSS). At this stage, the system ran for 30 days, during which there were cracks and leaks in the cylinder body, and the cultivation for activated sludge was interrupted for about 1 week.

### 3.3. The Sludge Growth Curve and Results Analysis

Adequate oxygen must be provided to maintain the metabolic activity of activated sludge. In the experiment, the input air is used as an oxygen supply method. The oxygen supply speed and oxygen supply quantity will directly affect the MLSS concentration. In order to make the activated sludge increase steadily, the aeration rate was stabilized from 1.6 to 2.1 m^3^/h.

The sludge was cultivated for about 40 d, with a temperature range of 11 to 23.5 °C, and a pH range of 6.8 to 8.7, which conformed to the requirements of the sludge cultivation. The sludge growth curve is shown in Figure 3.

At the initial stage of sludge cultivation, the microbial environment adaptability was poor. A higher aeration rate resulted in poor growth of microorganisms. As a whole, if the ratio of gas and water was controlled within the scope of 10:1–25:1, the aeration rate would have little influence on the growth of microorganisms. In the process of cultivation, considering the cost, the aeration rate was controlled at the lower limit. Because the designed concentration of the sludge was 3000 mg/L, too much time would be needed if using the complete stuffy exposure method to cultivate, and the test process would be affected. By adopting the combination of inoculation cultivation and the stuffy exposure method to cultivate sludge, the sludge concentration can meet the design value at a rapid speed within a short period of time.

## 4. Stable Experimental Section

### 4.1. Test and Analysis Methods

The pilot test lasted for 34 days in a continuously running mode with a hydraulic retention time of 6.5 h. In the pilot test, temperature, chromaticity, turbidity, pH, potassium dichromate index, NH_3_-N and the mixed liquor suspended solids (MLSS) were selected as the main indicators for the project, and their analytical methods mainly come from the references [44]. In brief, the chemical oxygen demand (COD) was measured by the dichromate reflux method. The NH_3_-N was measured by Nessler’s reagent colorimetry method. Turbidity was measured by the turbidimetric method. TChromaticity was determined by a platinum cobalt colorimetry. The sludge supernatant was obtained after filtering the fluid mixture samples through 0.45 μm membrane filters.

### 4.2. Results and Discussion

#### 4.2.1. The Effect of MBR Treatment on the Chromaticity of the Bath Wastewater

The effect of MBR treatment on the chromaticity of the bath wastewater is shown in Figure 4. The chromaticity of raw bath wastewater was 71–938 tu, which is large and unstable. However, the chromaticity decreased significantly in the effluent of the MBR treatment system. The average value of the chromaticity was 26.43 tu. The average removal rate of the chromaticity was 92.09%. The system was running continuously for 35 d. The value of the chromaticity was relatively stable in the effluent of the MBR treatment system. The value of the chromaticity did not decrease along with the increase in MLSS, which showed that the membrane filtration contributed greatly to the removal of chromaticity, but the effect of microorganisms was not obvious.

#### 4.2.2. The Effect of MBR Treatment on the Turbidity of the Bath Wastewater

The effect of MBR on the removal of the turbidity of the bath wastewater is shown in Figure 5. During the system operation, the water turbidity was controlled at 7.0–231.0 ntu, and it was unstable. After treatment, the turbidity of the treated water was stable at about 0.05 ntu, and the removal rate of the materials causing turbidity was above 99%. From the 5th day after the start of system operation, the water turbidity was constantly less than 0.05 ntu. During the operation, the raw water turbidity, water temperature, organic matter content and the change of operation conditions had no negative effects on the removal of turbidity. This could be interpreted by the screening function of the membrane that minimized the turbidity.

#### 4.2.3. The Effect of MBR Treatment on COD of the Bath Wastewater

If the composition of the organic matter in wastewater is relatively stable, the chemical oxygen demand (COD) and the biochemical oxygen demand (BOD) should follow a certain proportion. In general, the difference between the COD and the BOD of potassium dichromate at the first stage would be the organic matter content that cannot be microbially decomposed [45]. The COD removal of the bath wastewater by MBR is shown in Figure 6. The total COD removal rate of the system was 59–95%, with an average removal rate of 83%. The removal rate of the biofacies was 21–83%, with an average removal rate of 62%. After 12 d of continuous running, the COD of the membrane effluent was stable at a value under 50 mg/L.

As can be seen from Figure 6, the activated sludge system of the MBR played a main role in the removal of COD, and the membrane system had a stable effect on the removal of COD, compensating for the deficiency of the activated sludge system [46,47,48]. During the operation, although the COD value of the raw water was 155–562 mg/L, the MBR system had been relatively stable regarding the removal efficiency of COD. This suggests that the MBR system has a very strong ability to resist the impact of loading on COD removal. Because the membrane system intercepted the pollutants in the supernatant fluid, which would make the treatment inadequate for the sewage of biological systems, it improved the ability to resist the impact of loading on the MBR system at the same time, so as to ensure the stability of the effluent water quality of the MBR system [49,50,51]. Since the system began to run with an MLSS concentration above 3000 mg/L, and was not dredged during the test, this illustrated that the interception function of the MBR system not only ensured no sludge loss during the system operation, but also led to certain particularities when the system’s biology facies were compared with a general activated sludge system, making the activity and concentration of the activated sludge system always at a good level.

We also found that as the experiment progressed, the SRT extended continuously. The sludge concentration increased continuously, and the supernatant of the COD started to rise when reaching a minimum on the 25th d [52,53]. These could be attributed to two reasons. First, the F/M ratio was too low to sustain the growth needs of the microorganisms in the reactor, which led to a large number of microbes dying, and the degradation of the dead microbes was difficult, so the supernatant of COD increased. Second, the extension of SRT exacerbated endogenous respiration, producing a large number of soluble microbial products, which also led to an increase in the supernatant’s COD. However, the membrane module intercepted the microbial products with high molecular weight and kept them in the reactor, thus the content of COD was not high in the effluent of the MBR treatment system.

#### 4.2.4. The Effect of MBR Treatment on NH_3_–N Removal from the Bath Wastewater

The removal rate of NH_3_–N by the MBR was 44.9–100.0%, with an average of 72.2%. The biological removal rate was 28.8–100%, with an average of 65.2%. Because the ammonia nitrogen molecules were small, and cannot be intercepted by the membrane, the removal of NH_3_–N was mainly achieved by biodegradation. At the early stage of the test, the removal efficiency of NH_3_–N was minimal at a rate of 44.9–69.8%. As the test progressed, the removal efficiency of the NH_3_–N improved continuously, and the highest removal rate was 99%, attributed to the activated sludge intercepted by the MBR system that contained denitrifying bacteria [54], resulting in a good removal effect of NH_3_–N by the MBR system. After the system had been running for 20 d, the NH_3_–N of the membrane water was stable and under 10 mg/L (Figure 7).

#### 4.2.5. The Effect of MBR Treatment on LAS Removal from the Bath Wastewater

The removal rate of linear alkylbenzene sulfonates (LAS) by the MBR was 18.1–23.03%, with an average of 20.2%. The biological removal rate was 90.8–98.3%, with an average of 93.5%. As the test progressed, the LAS of the membrane water were stable and under 0.12 mg/L (Figure 8).

In theory, the removal of LAS by the bioreactor and membrane separation can be demonstrated by the difference between the concentration of LAS in the supernatant and the concentration of LAS in the treated water. The removal of LAS mainly depended on the biodegradation of the reactor, and the interception of the membrane made up for the instability of the biodegradation and ensured excellent effluent quality. The contribution of the MBR on removal mainly came from the adsorption of the membrane pore and membrane surface and the sieving adsorption of the membrane deposition layer.

## 5. Discussion on Costs and Benefits

### 5.1. Discussion on Costs

(1)Electricity. The power consumption was 40 KW/h-mo, corresponding to 0.55 CNY/(KW-h) or 0.73 CNY/m^3^.(2)Labor. The salary at the current rate is 180 CNY/day. As the project was tested in the laboratory, with a scale up to about 200 times the current lab scale, the cost would be about 0.90 CNY/m^3^.(3)Reagent cost. The concentration of sodium hypochlorite as an additive was 10 mg/L. With a price of 8 CNY/kg, the reagent cost would be about 0.08 CNY/m^3^.(4)Materials depreciation. The lifetime of the membrane could be up to five years, and the cost of the MBR membrane is CNY 600. Thus, the material cost of treating a cubic meter of water and the membrane replacement cost is CNY 0.33. The lifetime of the system equipment is about 10 years. Thus, the production of each cubic meter of water by the depreciation expense is CNY 0.22. The lifetime of the civil system is about 20 years. Thus, the depreciation expense of each cubic meter of water production is CNY 0.16.(5)The sum of the above costs is 2.42 CNY/m^3^. If mass produced, the cost will be even lower.

### 5.2. Comparison of Reuse Costs

Comparing the economic costs of different processes for bathing wastewater treatment, the MBR process can be used as a low-cost reuse treatment technology, taking into account that prices have increased over time, in Table 2.

## 6. Conclusions

Although the research on the treatment of bath wastewater by an MBR at the pilot scale was preliminary, the following conclusions can be drawn:(1)MBR is effective in treating bath wastewater. The effluent COD, NH_3_–N and LAS were <50, <10 and <0.12 mg/L, while the turbidity was <0.5 ntu and the chroma average was 26.4 tu. The quality of the effluent water meets the requirements of the urban miscellaneous water standard of China (GB/T 18920-2002) and conforms to the water reuse standard.(2)Although the COD and NH_3_–N of the treated water varied to some degree, the effluent was stable, suggesting that the sludge system of the MBR has very good ability to resist the impact of loading variation. The results suggest that MBR may also be used to treat sewage water whose water quality is inferior to the bath wastewater to meet the reuse standard.(3)Processing bath wastewater in the concentrated water area by MBR could reduce the cost of wastewater treatment operations and save water resources. As such, the economic benefits and social benefits are obvious.(4)Future studies should be focused on the following aspects. (1) Developing membrane materials with the capacity of high temperature resistance, contamination resistance, acid and alkali resistance properties and lowering the cost. (2) Developing membrane modules that can give full play to the membrane properties and can be developed on a large scale. (3) Determining the mechanism of membrane fouling, and finding the best way of prolonging the service life of the membrane. (4) Elucidating the mechanism of performance, considering the factors which influence the membrane separation process fully and reducing the parameters that need to be experimentally determined in the model. (5) Future development should also be combined with the use of various membrane separation technologies, the combination of membrane separation technology and conventional environmental processing units, higher separation performance and simple operation process systems.

## Figures and Tables

**Figure 1 membranes-11-00085-f001:**
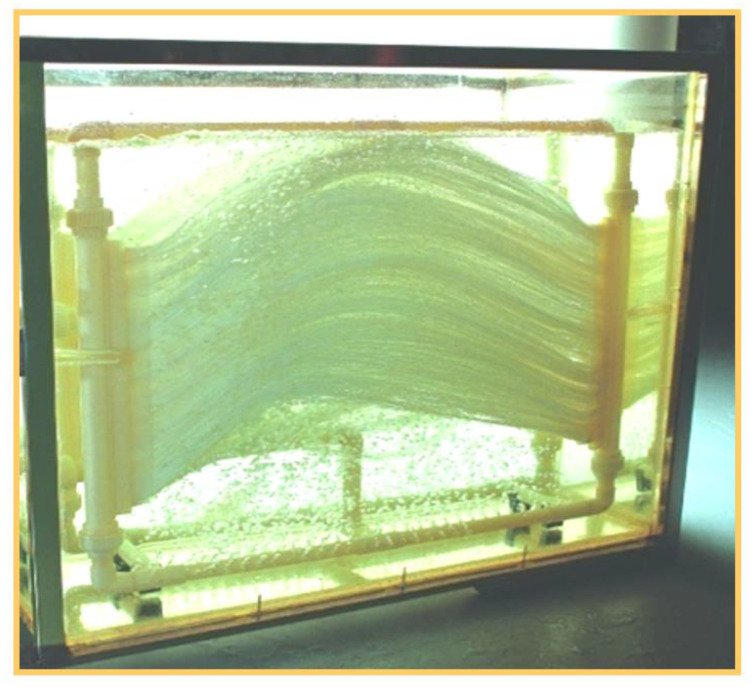
The device of the membrane bioreactor (MBR).

**Figure 2 membranes-11-00085-f002:**
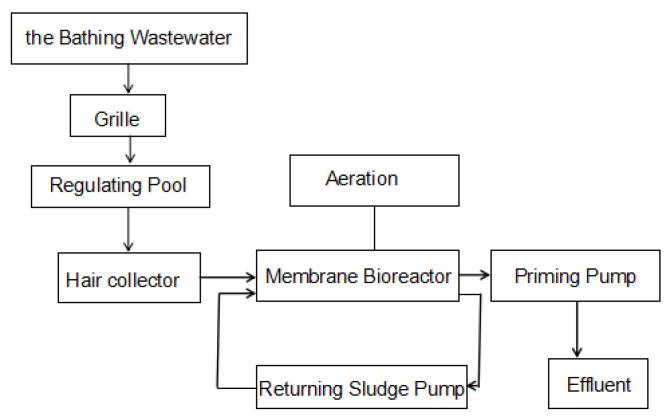
Process flow chart of the bath wastewater treatment by MBR.

**Figure 3 membranes-11-00085-f003:**
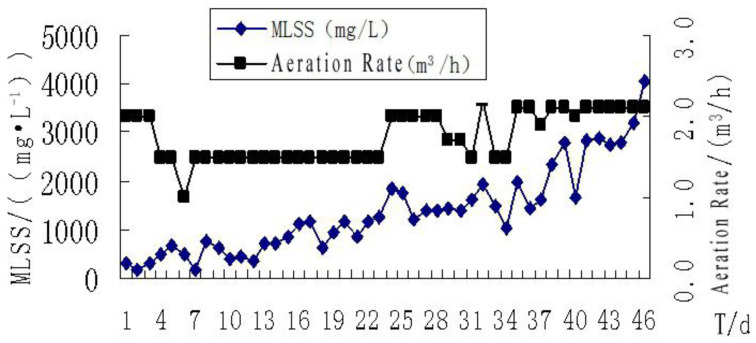
Growth curve of the activated sludge. MLSS—mixed liquor suspended solids.

**Figure 4 membranes-11-00085-f004:**
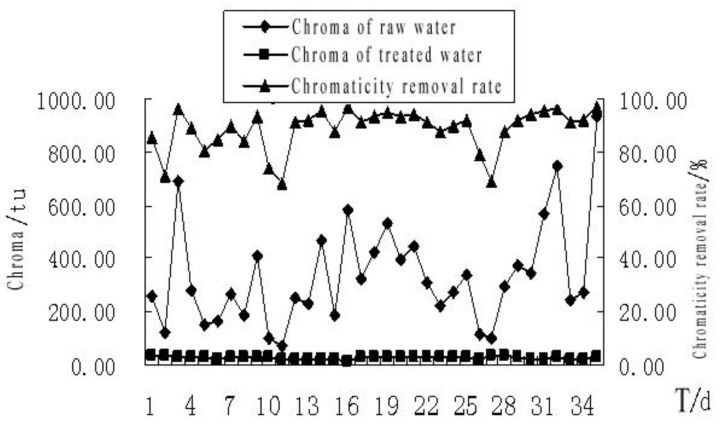
The effect of MBR treatment on the chromaticity of the bath wastewater.

**Figure 5 membranes-11-00085-f005:**
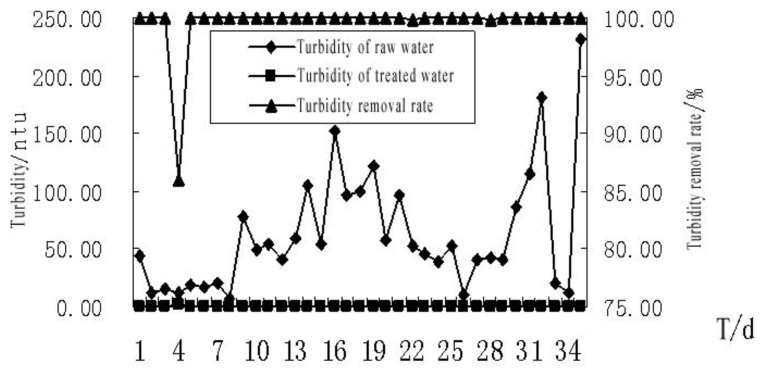
The effect of MBR treatment on the turbidity of the bath wastewater.

**Figure 6 membranes-11-00085-f006:**
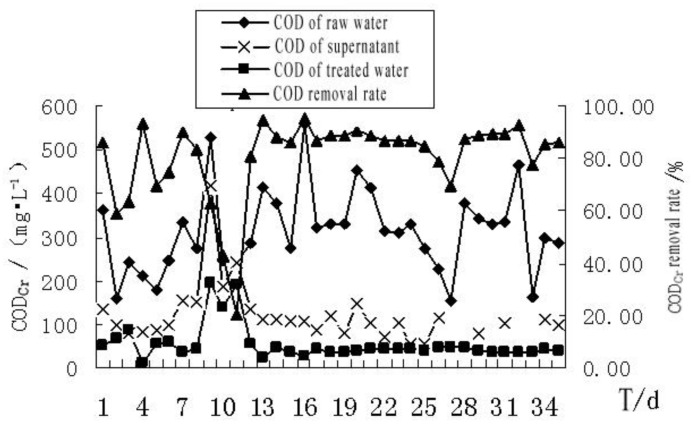
The effect of MBR treatment on COD of the bath wastewater.

**Figure 7 membranes-11-00085-f007:**
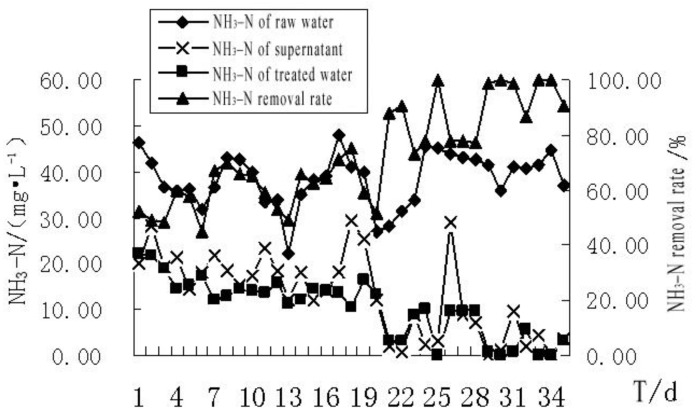
The effect of MBR treatment on the removal of NH_3_–N from the bath wastewater.

**Figure 8 membranes-11-00085-f008:**
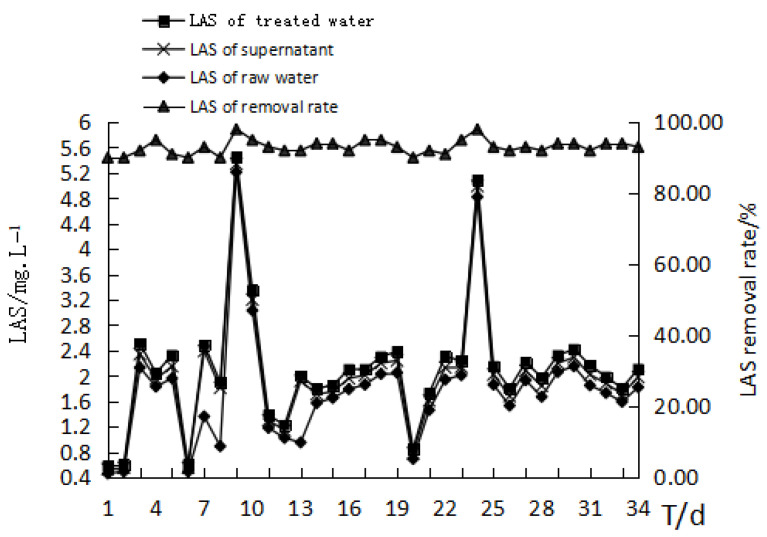
The effect of MBR treatment on the removal of LAS from the bath wastewater.

**Table 1 membranes-11-00085-t001:** Water quality of the bath wastewater.

Water Quality Index	Concentrations	Mean
COD (Chemical oxygen demand) (mg/L)	155–562	314
NH_3_–N (Ammonia nitrogen content index) (mg/L)	22.2–47.8	38.4
*Turbidity* (ntu)	7.0–231.0	62.1
*Chrominance* (tu)	71–938	334
LAS (Linear alkylbenzene sulfonates) (mg/L)	0.42–5.21	1.8
pH	7.3–8.5	7.9

**Table 2 membranes-11-00085-t002:** Comparison of reuse costs of bathing wastewater treated by different processes.

Reuse Cost (CNY/m^3^)	Processes	References	Notes
2.29	coagulation, filtration, adsorption	[14]	
1.61	submerged composite membrane bioreactor	[26]	
1.44	biological contact oxidization and membrane bioreactor	[27]	No depreciation charge
2.6	electrocoagulation–flotation	[28]	
2.42	MBR	this paper	

## Data Availability

Not applicable.

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
