# Peer review of "Pilot Tests on the Treatment of Bath Wastewater by a Membrane Bioreactor"

_membranes, 2021, doi:10.3390/membranes11020085_

Round 1

Reviewer 1 Report

The authors have improved the manuscript by following my suggestions for revising. I acknowldge the authors merits, but there is still some room for improving language, e.g. line 47-58.

Author Response

Thank you very much for your comments. We carefully and majorly revised the manuscript accordingly.

Reviewer 2 Report

The paper is devoted to up-to-date technique of wastewater treatment by means of membrane methods. The paper looks good, however, there are several significant drawbacks, which should be revised:

  1. Introduction section includes only 5 references. In my opinion introduction should rely on more sources to establish a problem to be solved.
  2. Authors chose only 5 parameters of raw water to be investigated, however if bath sewage is meant phosphorus should be also taken into account
  3. Authors only present figures for 34 days of experiment, however authors said that the experiment lasted for 5 months
  4. Figures, presented by the authors need to be more informative as they are now too small for deep analysis
  5. Authors only present dependencies of how the concentration drop after treatment, but it will be nice if authors can also demostrate the influence of different factors on it (MLSS, DO, etc.)
  6. Perhaps, the research and paper will look even better, if the experiment is divided into several parts - initial part and stable part, for example. As I understood from the paper, the results in the end of experiment were significantly better than in the beginning.
  7. Estimated costs can only refer to research of the authors, and can hardky be applied in real sector

Author Response

Comments from report #1

Thank you very much for your comments. We carefully and majorly revised the manuscript accordingly.

Comments from report #2

Phosphorus removal design is performed in front of the membrane module,So we did not consider the change of phosphorus before and after the membrane module treatment.

Comments from report #3

The membrane module was operated for 5 months.During the time, the experiments of treating high ammonia-nitrogen water ,domestic sewage and bathing wastewater were carried out.the experiments of treating bathing wastewater were carried out for 34 days.

Comments from report #4

Sorry, Fig. 2 is just a flow chart. No change is made.

Comments from report #5

Thank you very much for your comments. Because of some reasons, we can not supplement this part of experiment temporarily but we will consider to supplement later.

Comments from report #6

Thank you very much for your comments.I think you have a point. 

Comments from report #7

The cost of this test was only analyzed economically because no similar treatment water quality and process was found to be applied in real sector

Reviewer 3 Report

The manuscript by Yan et al. provides a case study on the treatment of bath wastewater by a membrane bioreactor (MBR). The manuscript provided brief literature review, experimental procedures, and results with cost analysis. However, I believe there are numerous areas where the manuscript could be improved. My comments are as follows:

1) Treatment of bath wastewater by a membrane bioreactor is a well-studied research area and has been done on a pilot-scale in several studies previously. There are studies on swimming center wastewaters in universities as well. The authors should explain the novelty of the study thoroughly.

2) The abstract should explain the novelty along with the results. Additionally, the authors should mention the issues associated with other purification technologies and how MBR can address those issues briefly.

3) Line 22: What’s the unit of time in “per time”?

4) Line no. 35: Please elaborate more on the types of MBR and the basis of the classification.

5) The introduction provides detailed information about membrane separation rather than focusing on MBRs, their advantages, previous studies and key results, issues in MBR treatments, novelty of this work, and how this work is potentially going to improve the understanding of MBR treatment for bath wastewater.

6) Line no. 47: Please provide reference for this line.

7) Line no. 50: I disagree with the authors on this statement. Membrane separation is considered one of the most economical separation processes.

8) Line no. 57: The acronym MBR is spelled incorrectly.

9) The authors should elaborate more on the selection of polypropylene as a membrane material. Was there a particular reason behind choosing it (such as robustness, compatibility, etc.)?

10) Please write the full forms of the terms or explain the concepts such as COD and NH3-N the first time they are mentioned.

11) Line no. 73: The authors have provided the test procedure in the form of a sketch. However, it needs to be thoroughly explained in the discussion as well. Additionally, the arrow directions at the ‘returning sludge pump’ in figure 2 are somewhat confusing.

12) Line no. 89-90: Please rephrase this sentence. Please explain the process and mention the important factors related to that process. Are the authors discussing about the important factors in the process of inoculation?

13) Section 2.4: I recommend that this section should be rewritten with a thorough explanation along with the process of inoculation, and the comparison with other sewage water. The authors should explain what the “second pond in the plant” refers to in line 96.

14) Line no. 114: Please rephrase the sentence as: but the cylinder cracked and leaked….

15) Line no. 127-129: This sentence is difficult co comprehend. Please rephrase or break it into two short sentences to make it easier to understand for the reader.

16) Section 3.1: The authors should elaborate more on the effect of aeration of sludge cultivation. It is not clear from the discussion provided at this point.

17) Line no. 186: Please rephrase the sentence as: “so the COD of effluent was not high.”

18) In the cost analysis, it would be helpful if the authors provide a comparison with the economics of other separation processes on the same scale and provide a perspective on the profitability of the MBR treatment.

19) Minor comment: There are several grammatical errors in the manuscript. Please proofread it thoroughly.

Author Response

Comments from report #1

Thank you very much for your comments. We carefully and majorly revised the manuscript accordingly.

Comments from report #2:

Thank you very much for your comments. We carefully and majorly revised the manuscript accordingly.

Comments from report #3

Thanks.“time” means  number of times

Comments from report #4

Thank you very much for your comments. We carefully and majorly revised the manuscript accordingly.

Comments from report #5

Thank you very much for your comments. We carefully and majorly revised the manuscript accordingly.

Comments from report #6

Thank you very much for your comments. We carefully and majorly revised the manuscript accordingly.

Comments from report #7

We agree with this comment.

Comments from report #8

Thank you very much for your comments. We carefully and majorly revised the manuscript accordingly.

Comments from report #9

Thank you very much for your comments. We carefully and majorly revised the manuscript accordingly.

Comments from report #10

Thank you very much for your comments. We carefully and majorly revised the manuscript accordingly.

Comments from report #11

Thank you very much for your comments. We carefully and majorly revised the manuscript accordingly.

Comments from report #12

The two principles of inoculating sludge, (1) bacteria were taken from sewage treatment plants with similar treatment processes, (2) bacteria were taken from sewage treatment plants with similar sewage properties. "Zhengzhou Wangxinzhuang Sewage Treatment" adopts the traditional activated sludge process.

Comments from report #13

“second pond in the plant”is  Secondary clarifier in the plant---Zhengzhou Wangxinzhuang Sewage Treatment

Comments from report #14

Thank you very much for your comments. We carefully and majorly revised the manuscript accordingly.

Comments from report #15

Thank you very much for your comments. We carefully and majorly revised the manuscript accordingly.

Comments from report #16

In this experiment, the method of sludge culture was the combination of inoculation culture and closed culture (I. E. Intermittent Culture) . Aeration was used to provide dissolved oxygen for the growth of microorganisms in sludge

Comments from report #17

Thank you very much for your comments. We carefully and majorly revised the manuscript accordingly.

Comments from report #18

The cost of this test was only analyzed economically because no similar treatment water quality and process was found

Comments from report #19

Thank you very much for your comments. We carefully and majorly revised the manuscript accordingly.

Round 2

Reviewer 2 Report

My comment from previous review considering figures was not considered, and figures remained less informative

Author Response

Point 1: Introduction section includes only 5 references. In my opinion introduction should rely on more sources to establish a problem to be solved.

Response 1: Thank you very much for your comments. We carefully and majorly revised the manuscript accordingly.

Point 2:Authors chose only 5 parameters of raw water to be investigated, however if bath sewage is meant phosphorus should be also taken into account

Response 2:The main pollutants in bathing wastewater are human skin secretion, grease, Dander, hair, dirt, synthetic detergent and perfume, as well as bacteria, fungi, E. coli and viruses. There was almost no phosphorus in the composition of these pollutants, so phosphorus contamination was not considered in the experiment. We added one parameter -LAS to be taken into account.

Point 3:Authors only present figures for 34 days of experiment, however authors said that the experiment lasted for 5 months

Response 3:The membrane module was operated for 5 months.During the time,the sludge retention time ,the experiments of treating high ammonia-nitrogen water ,domestic sewage and bathing wastewater were carried out.  

After sludge culture for 1 month, we studied the treatment of sewage after septic tank for 3 months, and after the system was stabilized, we studied the treatment of bathing wastewater.

The experiments of treating bathing wastewater were carried out for 34 days.

We carefully and majorly revised the time.To avoid suspicion, we only describe the treatment time for bathing wastewater as 34 days

Point 4:Figures, presented by the authors need to be more informative as they are now too small for deep analysis

Response 4:Thank you very much for your comments. We carefully revised figure2

Point 5:Authors only present dependencies of how the concentration drop after treatment, but it will be nice if authors can also demostrate the influence of different factors on it (MLSS, DO, etc.)

Response 5:Thank you very much for your comments. We add a discussion of the relationship between MLSS and DO.

Point 6:Perhaps, the research and paper will look even better, if the experiment is divided into several parts - initial part and stable part, for example. As I understood from the paper, the results in the end of experiment were significantly better than in the beginning.

Response 6:Thank you very much for your comments.I think you have a point. We carefully and majorly revised the manuscript accordingly.

Point 7:Estimated costs can only refer to research of the authors, and can hardky be applied in real sector

Response 7:Thank you very much for your comments.I think you have a point. We carefully and majorly revised the manuscript accordingly. 

Reviewer 3 Report

Thank you for providing the explanations to my questions and addressing my suggestions. However, the following questions from the first draft are not answered to my satisfaction.  

1) Treatment of bath wastewater by a membrane bioreactor is a well-studied research area and has been done on a pilot-scale in several studies previously. There are studies on swimming center wastewaters in universities as well. The authors should explain the novelty of the study thoroughly and add that explanation to the manuscript. It is very important to add the novelty of work.

2) The abstract should explain the novelty along with the results. Additionally, the authors should mention the issues associated with other purification technologies and how MBR can address those issues briefly. The abstract has not been modified in the updated manuscript. I strongly recommend that it needs to be updated.

3) Line no. 35: Please elaborate more on the types of MBR and the basis of the classification. No explanation has been added to this part of the manuscript.

4) The introduction provides detailed information about membrane separation rather than focusing on MBRs, their advantages, previous studies and key results, issues in MBR treatments, novelty of this work, and how this work is potentially going to improve the understanding of MBR treatment for bath wastewater. The manuscript still lacks more information on MBR. 

5) The authors have provided the test procedure in the form of a sketch. However, it needs to be thoroughly explained in the discussion as well. An explanation needs to be added for the test procedure instead of only showing it via a schematic. The reader should be guided step-by-step to really understand the test procedure.

6) Section 2.4: I recommend that this section should be rewritten with a thorough explanation along with the process of inoculation, and the comparison with other sewage water. 

7) Section 3.1: The authors should elaborate more on the effect of aeration of sludge cultivation. It is not clear from the discussion provided at this point. Please add this discussion to the manuscript.

Author Response

Point 1: Treatment of bath wastewater by a membrane bioreactor is a well-studied research area and has been done on a pilot-scale in several studies previously. There are studies on swimming center wastewaters in universities as well. The authors should explain the novelty of the study thoroughly.

Response 1:Thank you very much for your comments. The waste water quality of swimming center is different from that of bathing,and we intend to study the reuse of this kind of waste water.

Point 2:The abstract should explain the novelty along with the results. Additionally, the authors should mention the issues associated with other purification technologies and how MBR can address those issues briefly.

Response 2:Thank you very much for your comments. We carefully and majorly revised the manuscript accordingly.

Point 3:Line 22: What’s the unit of time in “per time”?

Response 3:Thanks.“time” means  number of times,We replace "per time" with "every time"

Point 4:Line no. 35: Please elaborate more on the types of MBR and the basis of the classification.

Response 4:Thank you very much for your comments. We carefully and majorly revised the manuscript accordingly.

Point 5:The introduction provides detailed information about membrane separation rather than focusing on MBRs, their advantages, previous studies and key results, issues in MBR treatments, novelty of this work, and how this work is potentially going to improve the understanding of MBR treatment for bath wastewater.

Response 5:Thank you very much for your comments. We carefully and majorly revised the manuscript accordingly.

Point 6:Line no. 47: Please provide reference for this line

Response 6:Thank you very much for your comments. We carefully and majorly revised the manuscript accordingly.

Point 7: Line no. 50: I disagree with the authors on this statement. Membrane separation is considered one of the most economical separation processes.

Response 7:Thank you very much for your comments.

I think you have a point.We carefully and majorly revised the manuscript accordingly.

Point 8:Line no. 57: The acronym MBR is spelled incorrectly.

Response 8:Thank you very much for your comments. We carefully and majorly revised the manuscript accordingly.

Point 9:The authors should elaborate more on the selection of polypropylene as a membrane material. Was there a particular reason behind choosing it (such as robustness, compatibility, etc.)?

Response 9:Thank you very much for your comments. We carefully and majorly revised the manuscript accordingly.

Point 10: Please write the full forms of the terms or explain the concepts such as COD and NH3-N the first time they are mentioned.

Response 10:Thank you very much for your comments. We carefully and majorly revised the manuscript accordingly.

Point 11: Line no. 73: The authors have provided the test procedure in the form of a sketch. However, it needs to be thoroughly explained in the discussion as well. Additionally, the arrow directions at the ‘returning sludge pump’ in figure 2 are somewhat confusing.

Response 11:Thank you very much for your comments. We carefully and majorly revised the manuscript accordingly.

Point 12:Line no. 89-90: Please rephrase this sentence. Please explain the process and mention the important factors related to that process. Are the authors discussing about the important factors in the process of inoculation?

Response 12:Thank you very much for your comments. We carefully and majorly revised the manuscript accordingly.

The oxygen supply speed and oxygen supply quantity will directly affect the MLSS concentration.In addition,the two principles of inoculating sludge, (1) bacteria were taken from sewage treatment plants with similar treatment processes, (2) bacteria were taken from sewage treatment plants with similar sewage properties. "Zhengzhou Wangxinzhuang Sewage Treatment" adopts the traditional activated sludge process.

Point 13: Section 2.4: I recommend that this section should be rewritten with a thorough explanation along with the process of inoculation, and the comparison with other sewage water. The authors should explain what the “second pond in the plant” refers to in line 96.

Response 13:Thank you very much for your comments. We carefully and majorly revised the manuscript accordingly.

“second pond in the plant”is  Secondary clarifier in the plant---Zhengzhou Wangxinzhuang Sewage Treatment

Point 14: Line no. 114: Please rephrase the sentence as: but the cylinder cracked and leaked….

Response 14:Thank you very much for your comments. We carefully and majorly revised the manuscript accordingly.

Point 15:Line no. 127-129: This sentence is difficult co comprehend. Please rephrase or break it into two short sentences to make it easier to understand for the reader.

Response 15:Thank you very much for your comments. We carefully and majorly revised the manuscript accordingly.

Point 16:Section 3.1: The authors should elaborate more on the effect of aeration of sludge cultivation. It is not clear from the discussion provided at this point.

Response 16:Thank you very much for your comments. We carefully and majorly revised the manuscript accordingly.

In this experiment, the method of sludge culture was the combination of inoculation culture and closed culture (I. E. Intermittent Culture) . Aeration was used to provide dissolved oxygen for the growth of microorganisms in sludge

Point 17: Line no. 186: Please rephrase the sentence as: “so the COD of effluent was not high.”

Response 17:Thank you very much for your comments. We carefully and majorly revised the manuscript accordingly.

Point 18:In the cost analysis, it would be helpful if the authors provide a comparison with the economics of other separation processes on the same scale and provide a perspective on the profitability of the MBR treatment.

Response 18:Thank you very much for your comments. We carefully and majorly revised the manuscript accordingly.

Point 19:Minor comment: There are several grammatical errors in the manuscript. Please proofread it thoroughly.

Response 19:Thank you very much for your comments. We carefully and majorly revised the manuscript accordingly.

Round 3

Reviewer 3 Report

I believe the authors have answered my questions and provided detailed discussions in the manuscript. I recommend that the manuscript should be published with a final proof-reading.

This manuscript is a resubmission of an earlier submission. The following is a list of the peer review reports and author responses from that submission.

Round 1

Reviewer 1 Report

The paper deals with an interesting practical application of membranes and considers the recent social changes occurring in China in the latest years. The work is scientifically sound. Some misprints are present and authors must read and arrange better the text. The first sentence is not clear. Authors claim that "water could be recycled..." but the investigation does not consider at all any biological parameter. Some reference to the law and to the purification targets required for this kind of tasks must be reported and critically discussed with respect to the results obtained by the authors. IN my opinion the paper can be published after these minor revisions. 

Reviewer 2 Report

The authors of the manuscripts entitled “Pilot Tests on the Treatment of Bath Wastewater by A Membrane Bioreactor”, have proposed a method towards reducing wasted water through water treatment and reuse. The manuscript suffers from many issues, the most important one being the written language. There are too many syntax and grammatical errors, and in many cases, it is not clear what the authors are trying to say. Regarding the proposed method, it is not clear what reuse the authors propose for the treated water. It is stated that the quality of the treated water surpasses the standards set for reuse, but these standards should be stated in the manuscript. Are the characteristics of the untreated water below the standards referred to by the authors? Finally, the disinfection stage should be incorporated in the process flow diagram and further discussed in the manuscript.

Reviewer 3 Report

The article presents results from a pilot test of an MBR for treatment of Bath wastewater. MBR treatment of bath wastewater seems to be a relevant technology for some applications. However, the novelty of the study is low, and the authors even fail to highlight and explain, what the aim of the study is. The introduction should be rewritten to actually explain the aim and the novelty of the study. In addition, there is a very low number of references in the introduction, and several statements are not supported with references.

There are several spelling errors that should be corrected.

Specific comments:

Line 19: what is meant with "... 0.4 tper adult per time"? 

Line 46: which university?

Table 1: what is the timespan of the pilot study? 

Line 53: "Acurtain box"???

Line 54: What is the membrane material? And is it a hollow fiber membrane?

Figure 2 should be more detalied

Section 2.3 should specify which methods are used to measure, e.g. NH3-N, etc. explain also which streams that are characterized.

Line 68: for what are these factors important, and why? 

What was tSRT and HRT? These are key parameters to understand your system. And was the flow though the system constant over time?

Line 79: Please explain what you mean by gas:water ratio

Line 84: 600-600???

Line 100-101: "A higher aeration rate resulted in poor growth..". How do you interpret this from Figure 3?

The price calculation is very simplified, not scalable and has low value, as a larger scale price calculation would look very different. 

Section 3 of the conclusion is too long, and some of the information could have been presented in the introduction.